# Alleviating Doctors’ Emotional Exhaustion through Sports Involvement during the COVID-19 Pandemic: The Mediating Roles of Regulatory Emotional Self-Efficacy and Perceived Stress

**DOI:** 10.3390/ijerph191811776

**Published:** 2022-09-18

**Authors:** Huilin Wang, Xiao Zheng, Yang Liu, Ziqing Xu, Jingyu Yang

**Affiliations:** 1School of Bussiness, Hunan University of Science and Technology, Xiangtan 411201, China; 2Faculty of Economics, Chulalongkorn University, Bangkok 10330, Thailand; 3International College, National Institute of Development Administration, Bangkok 10240, Thailand; 4Department of Medical Bioinformatics, University of Göttingen, 37077 Gottingen, Germany

**Keywords:** doctors, sports involvement, emotional exhaustion, self-efficacy, perceived stress, National Fitness Program

## Abstract

This study aims to understand the state of emotional exhaustion of Chinese doctors during the COVID-19 pandemic, and explore the role of sports involvement in enhancing doctors’ regulatory emotional self-efficacy, reducing stress perception, and alleviating emotional exhaustion. Finally, report the existing problems and make recommendations to the government and hospitals. The researchers constructed a cross-sectional questionnaire survey to collect data. From March to April 2022, using the snowball and convenience sampling methods, a total of 413 valid questionnaires were collected from 13 hospitals in Hunan Province. AMOS 23.0 was used to construct a structural equation model (SEM) with the bootstrapping approach to verify the proposed hypotheses. Doctors with more sports involvement exhibited higher levels of regulatory emotional self-efficacy and lesser perceived stress. Doctors who exhibited higher regulatory emotional self-efficacy had lesser perceived stress. The relationship between sports involvement and emotional exhaustion was mediated by perceived stress and/or regulatory emotional self-efficacy. Therefore, the government and hospitals should strengthen the depth and intensity of implementing the “National Fitness Program” at the hospital level, instead of just holding short-term activities with a small number of participants, but to cover all medical staff with fitness opportunities.

## 1. Introduction

Compared with the general population in China, Chinese doctors are under greater work pressure due to long-term heavy workloads, tense doctor-patient relationships, long overtime hours, difficulty in promotion, and work-life imbalance [1]. The dilemma faced by Chinese doctors is also what doctors in many countries, especially those in developing countries, are facing. Doctors are often frustrated at work, having to endure limited medical resources in their country or the local area, exaggerated media coverage of medical events, and distrust from patients in their ability to treat [2]. Under the weight of the COVID-19 pandemic and heavy workloads, doctors have had to take on a higher risk of infection and the uncertainty of being quarantined. A study from the 3rd Xiangya Hospital in Changsha, Hunan Province showed that only 9.5% of medical staff did not work overtime during the COVID-19 pandemic, and as high as 90% experienced overtime to varying degrees [3]. Another study found that more than 70 percent of Chinese doctors work more than 60 h a week, well above the legal limit of 44 h [4]. To perform their duties and serve society, in addition to being under greater psychological pressure than ordinary people, doctors have higher risks of anxiety, depression, insomnia, and other psychological problems than other groups [5,6]. During the COVID-19 pandemic, some doctors resigned or changed careers because they couldn’t handle the enormous pressure, and some doctors were sent to support the hardest-hit areas, which also meant that doctors who stayed on were under more stress than they were in the past.

Previous studies have shown that emotional exhaustion harms employees and their organizations, typically manifested as low work engagement, high absenteeism, low work efficiency, low job satisfaction, low organizational commitment, and poor performance [7]. Therefore, helping doctors alleviate emotional exhaustion during the COVID-19 pandemic is of great significance for doctors themselves, patients, hospitals, and society. Various types of doctors’ stress reduction programs have been used and studied, such as mindfulness-based interventions (MBIs), mind-body intervention (MBT-T), and cognitive behavioral therapy (CBT) program have been proven to effectively reduce stress, improve burnout, and enhance well-being [8,9,10]. To improve the physical and mental health of medical staff, more and more hospitals have begun to respond to the national strategic call for the “National Fitness Program (2021–2025)” [11], holding various sports activities and competitions, building sports facilities, and providing sports guidance. By participating in physical exercise, doctors can strengthen the body’s metabolism, release stress, relieve negative emotions, and generate positive self-adjustment.

Research on work engagement, burnout, emotional exhaustion, and mental health of health workers during the COVID-19 pandemic is currently receiving extensive attention [12]. However, existing studies mostly focus on the causes of job burnout, emotional exhaustion, and psychological problems in medical staff, and verify the effectiveness of psychosocial interventions in reducing doctors’ burnout and stress [13]. Different from previous studies focusing on psychological factors, this study is the first study to focus on how sports involvement can improve doctors’ regulatory emotional self-efficacy, reduce perceived stress, and relieve emotional exhaustion. This study fills the current research gap by linking physiological activities with psychological activities and, for the first time, quantitatively demonstrates the relationship between emotional regulation self-efficacy, perceived stress, and emotional exhaustion. Therefore, the objectives of this study are as follows: (1) To understand the emotional exhaustion of Chinese doctors during the COVID-19 pandemic; (2) To explore the effect of sports involvement on alleviating the emotional exhaustion of doctors; (3) To report the existing problems and make suggestions to the Hunan provincial government and hospitals.

Psychological interventions are limited in reducing occupational stress and burnout among doctors [12]. Based on job demands-resources (JD-R) theory, this study considers the path of physiological or behavioral interventions for relieving doctors’ emotional exhaustion, that is, strengthening physical exercise can effectively increase doctors’ self-efficacy in expressing positive emotions and managing negative emotions, and reduce the perception of stress, thereby alleviating emotional exhaustion. The researchers chose doctors in Hunan Province as the research subjects because Hunan is a province with a relatively advanced medical service capacity in China, especially with Xiangya Hospital, which is known as one of the best hospitals in China. Understanding the impact of sports involvement on doctors’ occupational stress and emotional exhaustion is not only helpful for the development of related theories, but also for the promotion of the “National Fitness Program (2021–2025)” at the hospital level.

## 2. Literature Review

### 2.1. Job Demands-Resources Theory

The job demands-resources model (JD-R model) is a popular conceptual model for work pressure and job burnout. Demerouti et al. [14] firstly published the paper in the Journal of Applied Psychology and formally proposed this conceptual framework. During the past 21 years, researchers have used this model to carry out relevant empirical studies in different fields, and the number of studies related to the model is increasing year by year [15]. JD-R model has been widely used in job burnout, organizational commitment, work engagement, etc. [16]. Furthermore, JD-R model has attracted the attention of occupational health agencies and government departments in countries around the world (especially in the United Kingdom, Europe, Canada, and Australia).

JD-R model is a very flexible theoretical framework in which various types of work environments and job characteristics can be classified into the following two categories: job demands and job resources. Job demands refer to the physical, psychological, social, or organizational requirements involved in the job. These demands require continuous physical or psychological effort or skills such as job stress, emotional demands, interpersonal demands workload, role ambiguity, physical demands, and work-family conflict [14]. Job resources refer to the material, psychological, social, or organizational resources that help individuals achieve work goals, reduce work requirements and related physical and mental consumption, and stimulate the individual’s personal growth, learning, and development. Includes job control, social support, feedback, compensation, career opportunities, organizational justice, etc. [17].

During the COVID-19 pandemic, hospitals have put forward a series of requirements for doctors’ work, such as complying with epidemic prevention and control regulations, doing a good job in patient treatment, doing scientific protection according to exposure risks, and participating in disease control training. However, insufficient resources are a typical job requirement [18]. Hospitals are demanding more from doctors than ever during the COVID-19 pandemic. When work requires excessive or insufficient resources, this unfair situation leads a doctor to perceive the resources and the energy loss, leading to the doctor often being in a state of fatigue, and generating strong pressure from work. In this work situation, the stress can lead to emotional exhaustion in the doctor, which seriously affects the doctor’s work behavior. However, the work stress perceived by doctors is due to the evaluation and feeling of individuals on the existing value of their organization and the serious lack of resources [19], that is, the serious imbalance between work demands and resources.

Many studies have preliminarily found that work resources provided in an organization can help individuals reduce the impact of work demands on work stress and play a buffer role [20]. As an invisible work resource, organizations often ignore overlook the role of sports involvement [21]. Many studies have shown that sports involvement has an essential impact on mental health [22], in which Downs and Strachan [23] measured subjects after physical exercise and showed a significant improvement in their regulatory emotional self-efficacy. In addition, some studies show that encouraging employees to participate in sports involvement can effectively increase employees’ positive emotions and thus reduce their perception of job stress [24]. This also means that when individuals have more work resources and face higher work demands, they will play a role in buffering emotional exhaustion.

Therefore, this study constructed an extended model based on the JD-R model and applied it to the study of emotional exhaustion among doctors during the COVID-19 pandemic. This also means that when an individual has more work resources and faces higher-intensity work demands, it will play a buffering role in emotional exhaustion.

### 2.2. The National Fitness Programs

In September 2020, President Xi Jinping proposed in his speech at a symposium of experts in the fields of education, culture, health, and sports that the government should focus on meeting the needs of the people, coordinate the construction of national fitness venues and facilities, and build a high-level national fitness public service system. As an important national development strategy, the “National Fitness Program” clearly pointed out that national health is an important manifestation of the country’s comprehensive strength and an important symbol of economic and social development and progress [25]. While the government has increased its financial investment in public sports facilities, it has encouraged society to widely carry out national fitness activities, and improve the physical health of the whole people by promoting sports participation [11].

The implementation of the “National Fitness Program” is one of the main means for China to respond to the COVID-19 pandemic crisis and implement active prevention and control [26]. Because sports activities can help people to vent their negative emotions reasonably and make people feel good physically and mentally [27]. It can also play a positive role in improving the quality of life to a certain extent [28]. Through sports involvement, people’s anxiety, depression, and even dissatisfaction can be transferred in the form of exercise [29]. This study also applies the variables of sports involvement to the doctor group, promotes national fitness to the special group of doctors, and analyzes how to reduce the perceived stress and emotional exhaustion of the doctor group during the COVID-19 pandemic.

## 3. Hypotheses

### 3.1. Sports Involvement and Regulatory Emotional Self-Efficacy

The current research has established a significant relationship between sports involvement and mental health [30]. Sports involvement can be explained as “an unobservable state of motivation, arousal, or interest” in viewing a game or participating in a sport-related activity that results in “searching, information-processing, and decision-making” [31]. Such sports involvement has the potential to motivate the individual’s happiness and physical fitness, it can help participants form an optimistic attitude and make them more confident. Regulatory emotional self-efficacy is an important variable of mental health. It refers to the confidence level of individuals in whether they can effectively manage their own emotional state. The study of the students shows that participation in physical activity is closely related to self-confidence, which can effectively improve students’ self-efficacy [32]. It showed that students participating in high school sports activities have a higher regulatory emotional self-efficacy [33]. Therefore, this study argues that physical exercise can enhance regulatory emotional self-efficacy. This study thus proposes hypothesis 1.

**Hypothesis** **1** **(H1).***Doctors with more sports involvement exhibit higher levels of regulatory emotional self-efficacy*.

### 3.2. Sports Involvement and Perceived Stress

The World Health Organization notes that during the COVID-19 pandemic, people have become more anxious, angry, stressed, agitated, and withdrawn [34]. Especially for the doctor group reported that a considerable proportion of healthcare workers had symptoms of depression (50.4%), anxiety (44.6%), insomnia (34.0%), and distress (71.5%) [35]. Zhang et al. [36] compared 927 medical health workers with nonmedical health workers and found that medical health workers had a higher prevalence of insomnia, anxiety, depression, somatization, and obsessive-compulsive symptoms. The literature exists showed that regular and sustained participation in sports involvement is associated with positive mental health [37]. It measured the samples after physical exercise and showed that their levels of anxiety, depression, tension, and stress were significantly reduced, and their levels of pleasure were significantly improved [38]. However, although physical activity is beneficial for mental health, the role sports involvement may play in mental health during the COVID-19 pandemic is currently not known. It is presumed that sports involvement will protect against poor mental health such as perceived stress for doctors, but to date, it has not been empirically investigated. Therefore, this study proposes Hypothesis 2.

**Hypothesis** **2** **(H2).***Doctors with more sports involvement have lesser perceived stress*.

### 3.3. Regulatory Emotional Self-Efficacy and Perceived Stress

At the same time, the research focuses on the relief of perceived stress by regulatory emotional self-efficacy. When an individual has a lower level of regulatory emotional self-efficacy, they cannot effectively regulate their various strong negative emotions, they may vent them in inappropriate ways, appearing in a state of rage, and more seriously, they will be affected by anxiety, depression, worry, and many other unfavorable factors [39]. When an individual has a higher level of regulatory emotional self-efficacy, they can control their emotions very well. Even in the face of financial difficulties, career failure, family discord, or academic failure, they can also get rid of negative emotions by adjusting their cognitive style, changing their original behavior, or seeking help from others [40]. Some past studies have shown that regulatory emotional self-efficacy is an important indicator of whether an individual can effectively manage their emotional state, and its role is precise to relieve tension, maintain emotional regulation, help regulate emotional impulses and promote mental health [41]. However, few studies have examined the regulatory emotional self-efficacy and how it impacts the perceived stress in the quantitative method, the current study aims at filling this gap, hence, this study proposed Hypothesis 3.

**Hypothesis** **3** **(H3).***Doctors who exhibit higher regulatory emotional self-efficacy have lesser perceived stress*.

### 3.4. The Mediating Effects

Regulatory emotional self-efficacy entails a subjective self-appraisal of one’s emotional competence in emotion regulation, and it reflects one’s confidence in own competence in emotion regulation [42]. Past research showed significant associations between regulatory emotional self-efficacy and mental health problems [43]. For example, studies have confirmed that regulatory emotional self-efficacy can significantly improve the effectiveness of individual anxiety and depression, and can directly predict depression and anxiety [44]. Regarding the association between sports involvement and regulatory emotional self-efficacy, there is some evidence that sports involvement can help improve emotions more positively [45], adolescent health [46], and participant sports can use for depression treatment [47]. It can be found that regulatory emotional self-efficacy played the mediating role between sports and mental health, however, most current research focus on the relationship between negative emotions [48], self-esteem [49], subjective well-being [44], and burnout [50]. In addition, emotional exhaustion impairs both personal and social functioning and often results in reduced work quality and damage to psychological health. Some studies evidence the relationship between stress and emotional exhaustion, for example, the study argued that for teachers at Higher Education Institutions total of 31.3% of the variance in emotional exhaustion was explained by perceived stress [51]. Despite the array of research foci, to the best of our knowledge, no study has examined the mediating role of regulatory emotional self-efficacy and perceived stress mediating the relationship between sports involvement and emotional exhaustion. Hence, this study proposed two mediating hypotheses as follows. This evidence will be important as doctors with huge stress and emotional exhaustion may require additional support as countries continue to manage the post-COVID-19 recovery phase.

**Hypothesis** **4** **(H4).**
*Regulatory emotional self-efficacy mediates the relationship between sports involvement and perceived stress.*


**Hypothesis** **5** **(H5).***Regulatory emotional self-efficacy and perceived stress mediate the relationship between sports involvement and emotional exhaustion*.

A summary of all hypotheses is shown in Figure 1.

## 4. Methods

### 4.1. Participants and Procedure

Snowball sampling and convenience sampling were used in this study. The researchers first contacted the heads of five hospital departments and asked them to distribute questionnaires to doctors, and got to know more hospital department heads under the introduction of these five heads. The researchers briefed all respondents about the purpose of the survey and informed them that the survey is anonymous and that all data are used for academic research only. Additionally, all doctors who participated in the survey were given a coffee voucher as a token of appreciation. From March 2022 to April 2022, 500 questionnaires were distributed to doctors in 13 hospitals in Hunan Province. After excluding invalid questionnaires, a total of 413 valid questionnaires were recovered, with a response rate of 82.6%.

Table 1 listed the demographic characteristics of the 413 doctors who participated in the survey. Among the respondents, (1) 43.8% were aged 29–44, followed by 39.5% aged 45–60. (2) In terms of gender, 54.7% of the respondents were male, and 45.3% of the respondents were female. (3) In terms of education level, 64.9% of the respondents had a bachelor’s degree, and 23.7% had a master’s or doctoral degree. (4) In terms of income, more than half of the respondents indicated that their monthly income was in the range of 10,000–20,000 CNY (1473–2946 USD). The demographic information of the survey was close to the data published in the China Health Statistical Yearbook, indicating that the sample was representative.

### 4.2. Measures

Perceived stress was measured by using 3 items of the Perceived Stress Scale developed by Cohen et al. [54], sample item includes “How often have you found that you could not cope with all the things that you had to do?”, and using a 5-point Likert scale ranging from 1 (i.e., never) to 5 (i.e., very frequently). To measure sports involvement, this study used the three items of the Sports Involvement Scale developed by Beaton et al. [55], sample item includes “Exercise plays an important role in my life”. The measure of regulatory emotional self-efficacy was derived from the Regulatory Emotional Self-Efficacy scale developed by Caprara et al. [42], sample item includes “I can manage negative feelings when reprimanded by someone important to me”. Emotional exhaustion was measured by combining parts of two scales (i.e., Maslach Burnout Inventory, Oldenburg Burnout Inventory) developed by Demerouti et al. [56], which were revised by Janurek et al. [57], sample item includes “I feel emotionally exhausted”. The above three scales used a 5-point Likert scale ranging from 1 (i.e., strongly disagree) to 5 (i.e., strongly agree).

### 4.3. Data Analysis

This study used AMOS 23.0 (IBM Corp, Armonk, NY, USA) to construct a structural equation model (SEM) to verify the proposed hypotheses. Following the suggestion of Anderson and Gerbing [58], this study took a two-step modeling approach: first, confirmatory factor analysis (CFA) was used to confirm the fit of the measurement model to the data, and then regression analysis and path analysis were performed on the structural model.

Considering that all data were generated by doctors’ self-reports, Harman’s one-factor test was used to examine the problem of common method variance (CMV) in the study. All items of the questionnaire were put into the factor analysis of SPSS, and the extraction factor was set as one. It can be concluded that the cumulative variance was 38.62%, so there is no CMV problem in this study [59].

## 5. Results

### 5.1. Measurement Model

Reliability analysis refers to measuring Cronbach’s alpha coefficients and composite reliability (CR) coefficients for latent variables [60]. Table 2 indicated that the Cronbach’α coefficients of all variables were in the range of 0.806–0.893, and the CR coefficients were in the range of 0.808–0.895, both of which were all higher than the recommended value of 0.7, indicating that all variables have good reliability [61]. Convergent validity is often used to measure whether each indicator reflects the same construct, including checking factor loadings and average variance extracted (AVE). According to the results in Table 2, the factor loading of all measurement items was between 0.706–0.904, and the AVE of all variables was between 0.585–0.682, both of which were higher than the recommended value of 0.5, indicating that all variables have high convergent validity [60]. The discriminant validity was checked by comparing the correlation coefficient of each variable with the square root of the AVE. As shown in Table 3, all correlation coefficients were less than the square root of AVE, proving that each variable has good discriminant validity.

### 5.2. Structural Path Model

The researchers first checked the error term and residual term of the structural model, and none of them showed negative values, indicating that the model complied with the basic fitness test criteria. The fit between the data and the structural model was high (χ^2^/df = 1.646, GFI = 0.964, AGFI = 0.946, NFI = 0.964, CFI = 0.985, TLI = 0.981, RMSEA = 0.040), much better than the values suggested by Hair et al. [61]. According to Table 3, there was a significant correlation among the independent, mediating, and dependent variables, which supported the validation of the hypotheses. The structural path model was presented in Figure 2. The effect of sports involvement on regulatory emotional self-efficacy was statistically significant (*β* = 0.447, *p* < 0.001), supporting H1; the effect of sports involvement on perceived stress was statistically significant (*β* = −0.269, *p* < 0.001), supporting H2; the effect of regulatory emotional self-efficacy on perceived stress was statistically significant (*β* = −0.316, *p* < 0.001), supporting H3.

The researchers hypothesized that sports involvement affects emotional exhaustion through two mediators, namely regulatory emotional self-efficacy and perceived stress. In this study, the bootstrap method was used to test the existence of mediating effects [62]. The results of the 5000-bootstrap sample at the 95% confidence interval were shown in Table 4: the absolute value of all Z values is greater than 1.96, and there is no zero value within the 95% confidence interval. Besides, regulatory emotional self-efficacy had a significant effect on the relationship between sports involvement and perceived stress (standardized indirect effect = −0.141, *p* < 0.001), supporting H4; regulatory emotional self-efficacy and perceived stress had a significant effect on the relationship between sports involvement and emotional exhaustion (standardized indirect effect = −0.287, *p* < 0.001), supporting H5. The findings suggested that doctors with higher sports involvement, higher regulatory emotional self-efficacy, and lower perceived stress exhibited lower levels of emotional exhaustion, and such doctors even may not have problems with emotional exhaustion.

## 6. Discussion

### 6.1. Contribution

This study discussed the impact of physical activity (i.e., sports involvement) on psychological factors (i.e., regulatory emotional self-efficacy, perceived stress, and emotional exhaustion) based on the JD-R theory. The results show that sports involvement had a significant positive impact on regulatory emotional self-efficacy; sports involvement had a significant negative impact on perceived stress; the impact of sports involvement on emotional exhaustion was mediated by perceived stress and/or regulatory emotional self-efficacy. Sports involvement had the greatest impact on emotional regulation self-efficacy, followed by the impact on perceived stress, and then alleviated emotional exhaustion through its impact on perceived stress. As shown in Figure 2, these variables from physical activity to mental activity explained 49% of the variance in emotional exhaustion, which is much higher than the 20–30% variance in previous studies [63]. The results showed that hospitals can not only relieve the emotional exhaustion of doctors through psychological intervention, but also through physical or behavioral intervention (i.e., sports involvement), which may be a more effective way. The results of this study promoted multidisciplinary applications in sports, medicine, and management, and enriched theories in related disciplines.

### 6.2. Practical Implications

Considering the doubling of workloads during the COVID-19 pandemic, doctors are more prone to emotional exhaustion than ever before. Therefore, considering the positive effects of enhanced sports involvement on reducing the perceived stress of individuals, improving their emotional regulation self-efficacy, and relieving individual emotional exhaustion, both the government and hospitals should increase the formulation of policies and measures to encourage doctors to exercise. The government can focus on emerging sports such as “cloud events” and “virtual sports”, relying on the national medical and health information service platform, to carry out a series of sports events jointly organized by hospitals across the country. Besides, the implementation of the policy also requires financial support. The government should set up special financial funds for hospitals, and use the funds to support hospitals to set up special performance bonuses for fitness, build supporting fitness equipment, and organize regular fitness courses. The government can also recruit fitness volunteers to improve fitness guidance and services for doctors by establishing a fitness volunteer service system.

The effectiveness of the “National Fitness Program” has been validated by academics. Zhang et al. [64] used panel data of 30 provinces/cities in China from 2008 to 2017, and the data analysis results showed that the “National Fitness Program” has a direct impact on national health. Besides, exercise is considered the preferred method to prevent different diseases, and the “National Fitness Program” plays a crucial role in the mitigation of non-communicable diseases [65]. However, most hospitals’ response and support for the “National Fitness Program” is only superficial. The hospital only held some simple events or activities, and such activities have a short cycle and few participants, and cannot achieve the purpose of national fitness at the hospital level. How to make doctors develop the habit of proper exercise is something that hospital managers should think about. For example, while the government is formulating policies, hospitals can use the medical and health information platform built by the government to form groups in various departments of the hospital. The total exercise volume and individual exercise volume of the group members should be counted weekly, so as to achieve the purpose of encouraging each other in the group and urging each other to do physical exercise. In addition, the hospital can offer a set of sports elective courses every six months for medical staff to choose from. Hospitals should create good sports conditions for medical staff with regular courses, advanced facilities, and professional coaches.

## 7. Conclusions

In response to the proposed research objectives, this study stated that about 75% of doctors in Hunan Province experienced varying degrees of emotional exhaustion during the COVID-19 pandemic. Besides, the results showed that doctors with higher sports involvement had better regulatory emotional self-efficacy, lesser perceived stress, and lesser emotional exhaustion. Therefore, this study suggested that the government and hospitals should strengthen the depth and intensity of implementing the “National Fitness Program” at the hospital level, instead of just holding short-term activities with a small number of participants, but to cover all medical staff with fitness opportunities.

This study has certain limitations. First, the survey only covers the situation of doctors working in cities in Hunan Province. Future research should try to compare the different conditions of healthcare workers in urban and rural areas. Second, this study is a cross-sectional study, and future research should be considered in the form of a longitudinal study, which can more accurately and effectively reflect the impact of sports involvement on the physical and mental health of medical staff. Third, the Sports Involvement Scale used in this study has a certain error in accurately reflecting the degree of sports involvement of the respondents. The scale contains psychological and physical factors, and the researchers believe that psychological factors should be excluded. Therefore, follow-up research should use a scale that can more accurately reflect the respondents’ sports involvement, and combine them with longitudinal research methods, such as asking respondents how often they participated in sports in the past week.

## Figures and Tables

**Figure 1 ijerph-19-11776-f001:**
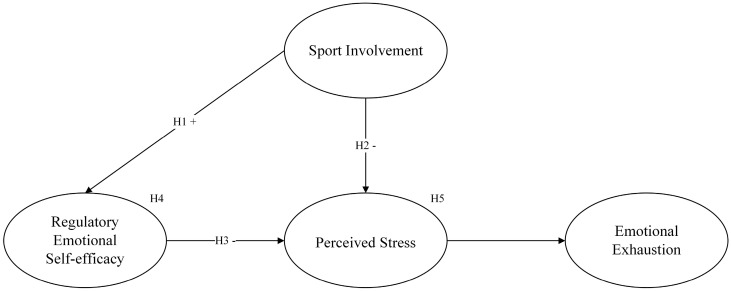
The hypothesized model.

**Figure 2 ijerph-19-11776-f002:**
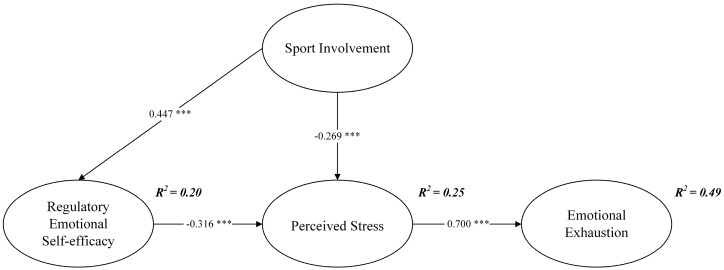
Structural path model. *** *p* < 0.001. Standardized coefficients are reported.

**Table 1 ijerph-19-11776-t001:** Participant profile (N = 413).

Profiles	Survey (%)	2020 Statistical Yearbook Estimates ^a^
** *Respondent age (%)* **		
≤28	21.3	25–34 (28.0%)
29–44	43.8	35–44 (33.3%)
45–60	34.9	45–54 (22.0%) 55–59 (7.9%) ≥60 (8.8%)
** *Respondent gender (%)* **		
Male	54.7	54%
Female	45.3	46%
** *Respondent education level (%)* **		
Higher vocational certificate	11.4	15.8.%
College/University	64.9	55.9%
Master or Ph.D.	23.7	22.7%
** *Monthly salary* **		
≤5000 CNY	11.6	Mean 8504 CNY ^b^
5001–10,000 CNY	33.2	
10,001–20,000 CNY	51.8	
≥20,001 CNY	3.4	

^a^ National Health Commission [52], ^b^ Hunan Provincial Bureau of Statistics [53].

**Table 2 ijerph-19-11776-t002:** Reliability and validity test.

Items	Loadings	Cα	AVE	CR
** *Sport Involvement (SI)* **		0.806	0.585	0.808
SI1	0.794			
SI2	0.706			
SI3	0.792			
** *Regulatory Emotional Self-efficacy (RES)* **		0.893	0.682	0.895
RES1	0.777			
RES2	0.904			
RES3	0.769			
RES4	0.846			
** *Perceived Stress (PS)* **		0.820	0.608	0.823
PS1	0.776			
PS2	0.824			
PS3	0.736			
** *Emotional Exhaustion (EE)* **		0.846	0.649	0.847
EE1	0.764			
EE2	0.822			
EE3	0.828			

**Table 3 ijerph-19-11776-t003:** Discriminant validity test.

Construct	PA	RES	PS	EE
SI	**(0.765)**			
RES	0.393 **	**(0.826)**		
PS	−0.331 **	−0.380 **	**(0.780)**	
EE	−0.302 **	−0.307 **	0.584 **	**(0.806)**

The square root of the average various extracted (AVE) is in diagonals (bold); off diagonals are a Person’s corrections of contracts. ** *p* < 0.01.

**Table 4 ijerph-19-11776-t004:** Standardized direct, indirect, and total effects.

	Point Estimate	Product of Coefficients		Bootstrapping				
				Percentile 95% CI		Bias-Corrected 95% CI		Two-Tailed Significance
		*SE*	*Z*	Lower	Upper	Lower	Upper	
** *Direct effects* **								
SI → RES	0.447	0.059	7.576	0.330	0.561	0.331	0.562	0.000 (***)
SI → PS	−0.269	0.066	−4.076	−0.399	−0.141	−0.398	−0.140	0.000 (***)
RES → PS	−0.316	0.060	−5.267	−0.435	−0.196	−0.436	−0.198	0.000 (***)
PS → EE	0.700	0.041	17.073	0.616	0.779	0.614	0.777	0.000 (***)
** *Indirect effects* **								
SI → PS	−0.141	0.032	−4.406	−0.209	−0.084	−0.216	−0.088	0.000 (***)
SI → EE	−0.287	0.046	−6.239	−0.379	−0.199	−0.379	−0.199	0.000 (***)
RES → EE	−0.221	0.044	−5.023	−0.306	−0.138	−0.308	−0.138	0.000 (***)
** *Total effects* **								
SI → RES	0.447	0.059	7.576	0.330	0.561	0.331	0.562	0.000 (***)
SI → PS	−0.410	0.056	−7.321	−0.515	−0.300	−0.515	−0.298	0.000 (***)
SI → EE	−0.287	0.046	−6.239	−0.379	−0.199	−0.379	−0.199	0.000 (***)
RES → PS	−0.316	0.060	−5.267	−0.435	−0.196	−0.436	−0.198	0.000 (***)
RES → EE	−0.221	0.044	−5.023	−0.306	−0.138	−0.308	−0.138	0.000 (***)

Standardized estimations of 5000 bootstrap samples. *** *p* < 0.001.

## Data Availability

Not applicable.

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
