# Peer review of "Alleviating Doctors’ Emotional Exhaustion through Sports Involvement during the COVID-19 Pandemic: The Mediating Roles of Regulatory Emotional Self-Efficacy and Perceived Stress"

_ijerph, 2022, doi:10.3390/ijerph191811776_

Round 1
Reviewer 1 Report
Thank you to the authors for the opportunity to read their article. Below are my comments and suggestions.
Abstract:
In the abstract it is better to specify what research tools were used than the statistical program
Please add keywords: self-efficacy, perceived stress
Introduction
In this part, it is worth adding the information that doctors remaining at work are under much higher pressure due to the absence of other doctors during the COVID-19 pandemic.
Does the author know the amount of overtime hours doctors must perform in their work? Please complete this. https://doi.org/10.3390/ijerph16173049
Have the authors found any other studies describing physicians' stress reduction programs during the covid-19 pandemic? How effective were these programs?
2. Literature review
Line 87 ”work enjoyment” Shouldn't there be: ”work engagement”
Line 101 When presenting the JD-R model, the authors should describe in more detail the requirements in the workplace for doctors, especially during the COVID-19 pandemic.
In the theoretical part, one or two paragraphs should be devoted to the role of regulators emotional, physical activity, and self-efficacy in reducing stress.
3. Hypotheses
This section is understandable and well written, no changes needed. One note: please indicate the potential impact (- or +) on figure 1
4. Methods
4.2. Measures
Descriptions of research tools should include a) the name of the questionnaire, b) an example question.
5. Results
The authors should consider whether the data in Table 2 should not only shorten the information on the reliability of Cronbach's alpha in section "4.2. Measures"
The information in this section is sufficient
6. Discussion
Do the authors know of any other studies on the effectiveness of the "National Fitness Program (2021-2025)"?
Reviewer 2 Report
The paper deals with finding the explanation how the sports involvements reduce the doctors’ emotional exhaustion. The authors conclude that RES and PS are mediators. The final suggestion of the authors is: to increase the sport facilities.
Although the method is appropriate, some questions arise.
Based on the questionnaires, sports involvement means not only doing but discussing sports.
It would be interesting whether to what extent the sports activity (SI1, SI4 and SI5) is responsible for the effect and to what extent thinking of sport? I think that SI6 is rather psychological than physical factor. So are SI2, SI3.The authors consider SI1-SI6 as physical factor.
Being fan, it is as good as running? (It is not excluded, it is a kind of relaxion and it may decrease the stress in the work.)
Why the author think that SI does not affect EE directly?
Figure 2 is missing.
What is PA in Table 3?
Instead of RES, often ERS is written in Table 4.
Table 1: Monthly salary in Yearbook was omitted.
What does it mean: „CMV was not considered to be a serious threat to this study”? Medium threat?
I suggest accepting the paper after major revision.
Round 2
Reviewer 2 Report
The paper became better, I suggest its acceptation